# Studying Brain Activation during Skill Acquisition via Robot-Assisted Surgery Training

**DOI:** 10.3390/brainsci11070937

**Published:** 2021-07-16

**Authors:** Kurtulus Izzetoglu, Mehmet Emin Aksoy, Atahan Agrali, Dilek Kitapcioglu, Mete Gungor, Aysun Simsek

**Affiliations:** 1School of Biomedical Engineering, Science and Health Systems, Drexel University, Philadelphia, PA 19104, USA; 2School of Education, Drexel University, Philadelphia, PA 19104, USA; 3Center of Advanced Simulation and Education (CASE), Acibadem Mehmet Ali Aydinlar University, Istanbul 34684, Turkey; Emin.Aksoy@acibadem.edu.tr (M.E.A.); dilekitapci@gmail.com (D.K.); 4Department of Biomedical Device Technology, Acibadem Mehmet Ali Aydinlar University, Istanbul 34684, Turkey; atahanagrali@gmail.com; 5Obstetrics and Gynecology Department, Medical School, Acibadem Mehmet Ali Aydinlar University, Istanbul 34684, Turkey; mtgungor@gmail.com; 6Department of General Surgery, Camlica Medipol University Hospital, Istanbul 34214, Turkey; aysun.simsek@medipol.edu.tr

**Keywords:** functional near-infrared spectroscopy, neuroimaging, fNIRS, clinical skill acquisition, robot-assisted surgery, simulation-based training

## Abstract

Robot-assisted surgery systems are a recent breakthrough in minimally invasive surgeries, offering numerous benefits to both patients and surgeons including, but not limited to, greater visualization of the operation site, greater precision during operation and shorter hospitalization times. Training on robot-assisted surgery (RAS) systems begins with the use of high-fidelity simulators. Hence, the increasing demand of employing RAS systems has led to a rise in using RAS simulators to train medical doctors. The aim of this study was to investigate the brain activity changes elicited during the skill acquisition of resident surgeons by measuring hemodynamic changes from the prefrontal cortex area via a neuroimaging sensor, namely, functional near-infrared spectroscopy (fNIRS). Twenty-four participants, who are resident medical doctors affiliated with different surgery departments, underwent an RAS simulator training during this study and completed the sponge suturing tasks at three different difficulty levels in two consecutive sessions/blocks. The results reveal that cortical oxygenation changes in the prefrontal cortex were significantly lower during the second training session (Block 2) compared to the initial training session (Block 1) (*p* < 0.05).

## 1. Introduction

Robot-assisted surgery (RAS) systems have changed the way surgeons operate and teach surgery in the last two decades [1]. RAS systems have enhanced several surgical techniques and are now widely used in the surgical disciplines including urology, general surgery, gynecology, cardiovascular surgery, endocrine surgery and thoracic surgery [1,2,3,4]. As such robotic systems provide numerous benefits for both patients and surgeons, an exponential increase in surgical operations has been expected, and just in 2015, more than 650,000 surgical procedures had been performed by using RAS all over the world [5]. Among many other advantages, less postoperative pain due to smaller incisions, less risk of postoperative infections and shorter hospitalization periods and recovery times are the main benefits for the patient. By performing surgical procedures through robot-assisted surgery, surgeons have the following advantages: high-quality vision of the surgical site due to ten times magnification, better hand–eye coordination compared to laparoscopic surgery, less risk of tremor, more comfort due to a better ergonomic position and having a high precision of motion in terms of speed and range of the motion. On the other hand, this increasing trend of use in various surgical disciplines creates new challenges and requirements, particularly in the area of teaching and training, that need to be studied and addressed for the proper use and efficacy of RAS systems [6]. 

The essential parts of RAS training are to understand robotic technology, become familiar with the system itself and the device functions, gain a basic knowledge of troubleshooting during the operation and realize the limitations of the system [6]. To ensure the safety of the patient during robot-assisted surgical procedures, trainees must practice in a safe environment to improve their basic robotic skills, and to perform complex maneuvers. There are different training courses such as the Robotic Training Network (RTN), the Fundamentals of Robotic Surgery (FRS) Program, the Fundamental Skills of Robotic Surgery (FSRS) Program and the Morristown Protocol [7,8,9]. FRS is a widely used course for robotic surgery training, which consists of a multi-specialty, proficiency-based curriculum of basic technical skills to train surgeons and assess their performance [7,9]. 

The aforementioned clinical programs have become more common training sessions with the simulator-based metric that the robotic simulator tracks for each task to be completed. Besides complementing these performance-tracking systems and metrics, brain-based measures via wearable functional brain imaging systems can also enable us to better understand and evaluate trainees’ performance [10]. Hence, the aim of this research was to study brain activation through measures of the changes in hemodynamic responses from the cortical regions associated with attention and working memory while trainees were engaged with robot-assisted surgery training. The central hypothesis is that the cortical responses, i.e., mental workload changes, are correlated with varying task difficulty levels. Studying these measures and the correlation with a simulator-based metric and brain-based assessment techniques will help enhance these RAS simulators’ performance assessment methods. 

Nowadays, neuroimaging techniques including electroencephalography (EEG), functional magnetic resonance imaging (fMRI), magnetoencephalography (MEG), positron emission tomography (PET) and functional near-infrared spectroscopy (fNIRS) are the candidates for such performance assessment scoring systems. Due to its advantages of being noninvasive, wearable and easy to use for measures from the prefrontal cortex in field settings, fNIRS was utilized as the neuroimaging technique for this research study. 

fNIRS is an optical system which utilizes near-infrared (NIR) light in brain activity assessment by measuring light absorption differences in the changes in oxy-hemoglobin and deoxy-hemoglobin concentrations [11]. Light-emitting diodes (LED) are used as the light source, and photons interacting with the brain through tissue absorption and scattering are acquired via photodiodes. The modified Beer–Lambert law is used to calculate and convert the light intensity measures acquired from the brain to hemodynamic changes [12,13]. Hemodynamic responses in the prefrontal cortex (PFC) region of the brain elicited by different stimuli can be monitored with these optical systems [14,15,16,17]. Further, fNIRS has been used as an assessment tool in medical simulation training in addition to the current scoring systems [18,19]. The objective of this study was to investigate whether neurophysiological measures of trainees could be used as an additional monitoring tool for assessing the training effect of robotic surgery simulators. Hence, the experimental protocol focused on acquiring hemodynamic changes via fNIRS from the prefrontal cortex regions of the trainees during robotic surgery simulation sessions.

## 2. Materials and Methods

### 2.1. Participants

Twenty-four surgery residents between the ages of 26 and 32 volunteered to participate in this study (mean age ± SD = 28.25 ± 1.98 years, 18 females, 6 males). Nineteen of the participants were from an obstetrics and gynecology (OB&GYN) department, and the remaining five residents were from a general surgery department. Prior to the study, all the participants agreed and signed written informed consent for voluntary participation in the study. The informed consent was reviewed and approved by the Ethical Committee of Acıbadem Mehmet Ali Aydinlar University, Istanbul, Turkey.

### 2.2. Experimental Protocol

All participants had a familiarization session, before the actual experimental protocol began. During the familiarization session, an expert RAS instructor not only introduced the participants to the components of the simulator system but also demonstrated the RAS system (Da Vinci Surgical System Xi console with backpack simulation module: Intuitive Surgical Inc., Sunnyvale, CA, USA). The task and RAS familiarization session lasted between 10 and 12 min, and the following topics were covered: surgeon console adjustments, robotic arms and clutch control, zoom in and out, camera targeting. After the familiarization session, the participants performed a ring walk exercise at the easiest level with aids from the instructor. The goal for this short tutorial session was to acclimate the participants to the RAS simulator as much as possible. After an fNIRS sensor headband was placed on the forehead, the participants were instructed to begin the actual training protocol. 

Each participant completed two blocks of sponge suturing tasks. Both blocks consisted of the same three tasks. In Block 1, tasks were presented in difficulty order, starting with the easiest task and ending with the hardest task. Block 2 presented the same tasks in a randomized order (Figure 1).

Robot-Assisted Surgery (RAS) Simulator. In this study, we utilized the Da Vinci Surgical System Xi console (Intuitive Surgical Inc., Sunnyvale, CA, USA) as the RAS simulator. The simulator is an attachment to the surgeon console of the Da Vinci Surgical System. The RAS simulator has a simulator-based metric (embedded scoring system), and we utilized the simulator’s logged scores as the assessment for each task performance. The simulator-based metric of the simulator evaluates the participants based on various performance metrics, as provided in Table 1. This RAS system combines these metrics and generates a composite score to evaluate the overall performance of the trainee surgeons. The range of scores varies from a maximum of 1440 to a minimum of 0 (zero) points.

Surgical Tasks. All participants completed a total of six suture sponge tasks in two blocks. The system did not allow participants to start the next task before finishing the task at hand. On all suture sponge tasks, all the participants were asked to insert and extract the needle through several pairs of targets located at the edge of the sponge. The variations in the target positions and operations with the console increased with the difficulty of the levels (see Figure 2). 

Functional Near-Infrared Spectroscopy. A continuous-wave fNIRS system, Imager 1200 (fNIR Devices LLC., Potomac, MD, USA), was utilized to monitor and record brain activity changes from the prefrontal cortices. The fNIRS sensor headband provides 16-optode scans at a sampling rate of 2 Hz from the PFC region by employing light sources with 730 nm and 850 nm wavelengths (four LEDs) and detector (ten photodetectors) pairs (Figure 3b).

### 2.3. Data Processing and Analysis

The trainee surgeons’ performance data during RAS training consist of the cortical oxygenation changes from the prefrontal cortex acquired via fNIRS, proficiency scores assigned by the simulator and the times recorded for each task completion. 

fNIRS Data. The non-cortical signals, mainly systemic changes which have a higher frequency than the hemodynamic changes, should be teased out; hence, a low-pass finite impulse response filter was applied with a 0.1 Hz cut-off frequency to the raw light intensity data. Raw data from each optode (Figure 3b) were examined for saturated and noisy signals for the manual optode rejection. A sliding window motion artifact rejection filter was applied to exclude artifacts caused by the head motions from the data [20]. The fNIRS data then were processed by applying the modified Beer–Lambert law to convert the raw light intensity measures to hemodynamic changes, i.e., oxygenated hemoglobin (HbO) and deoxygenated hemoglobin (HbR). The biomarker of interest in this study was calculated using values of HbO and HbR for each optode from the PFC region, namely, oxygenation (Oxy = HbO − HbR). The fNIRS measures of HbO, HbR and Oxy were included in the analysis for each PFC quadrant, namely, left dorsolateral PFC (DLPFC), left anterior medial PFC (AMPFC), right AMPFC and right DLPFC. Corresponding optode locations for each quadrant are as follows: left DLPFC (optodes 1–4); left AMPFC (optodes 5–8); right AMPFC (optodes 9–12); right DLPFC (optodes 13–16).

Statistical Analysis. Linear mixed effect (LME) models allow additional control for variation between subjects, and mitigating errors resulting from missing data and unbalanced groupings [21,22]. Hence, statistical analyses were conducted using R (ver. 3.6.1) by employing the lme4 [23] and lmerTest [24] packages in R for construction and evaluation of LME models. LME models were used to investigate the main and interaction effects of training blocks (Block 1, Block 2) and task difficulty levels (easy, medium and hard) on behavioral (score and task completion time) and fNIRS (HbO, HbR and Oxy from left DLPFC, left AMPFC, right AMPFC and right DLPFC) measures. The significance of fixed effect terms was evaluated using likelihood ratio tests, where the full effects model was compared against a model without the effect in question (e.g., 1 + Block + (1|Subject) vs. 1 + (1|Subject) when evaluating a significant main effect of Block). Post hoc analysis was conducted to evaluate differences between levels per model term. A total of twelve planned post hoc comparisons were performed. Three comparisons were between task difficulty levels (e.g., easy vs. hard), three were between blocks of the same difficulty level (e.g., Block 1 vs. Block 2 of easy) and six were between task difficulty levels per block (e.g., easy vs. hard in Block 1). Descriptive statistics are presented as mean ± standard deviation (Table A1, Table A2 and Table A3). Homogeneity of variance, and normality of residuals and random effects were assessed using visual inspection. If model predictions showed heteroscedasticity or a non-normal distribution, then log10 transformations were performed on the response variables. Satterthwaite approximation of degrees of freedom was used in post hoc analyses. For all statistical analyses, the level of significance was set at α = 0.05. Adjustments using the false discovery rate (FDR) were made on *p*-values to account for Type I error inflation. Cohen’s d was used to examine post hoc effects. A d of 0.2 was considered a small effect, while 0.5 and 0.8 represent medium and large effects, respectively.

## 3. Results

The main effects between participants’ task performance (simulation-assigned scores and task completion times) and on participants’ hemodynamic responses (Oxy, HbO, HbR) across two blocks and task difficulty levels were investigated. As it is depicted in Figure 1, the tasks were presented in difficulty order, i.e., starting with the easiest task and ending with the hardest task in Block 1, whereas the same tasks were administered in a randomized order during Block 2.

Behavioral Results. The main effect of block and task difficulty was significant for score (χ^2^(1) = 21.92, *p* < 0.001; χ^2^(2) = 27.52, *p* < 0.001), and task completion time (χ^2^(1) = 114.37, *p* < 0.001; χ^2^(2) = 6.70, *p* = 0.035). These changes are also shown in Figure 4 for both behavioral measures. Post hoc testing between task difficulty levels indicated significant decreases in score from easy to hard (*p* < 0.001, d = 1.10) and medium to hard (*p* < 0.001, d = 0.89), and increases in completion time from medium to hard (*p* < 0.022, d = −0.59). The effect of the interaction between block and task difficulty was significant for both score (χ^2^(2) = 11.37, *p* = 0.003), and task completion time (χ^2^(2) = 28.68, *p* < 0.001). Post hoc testing of the same difficulty levels across blocks revealed a significantly higher score in Block 2 than in Block 1 for the easy (*p* < 0.001, d = −1.72) and medium (*p* = 0.023, d = −0.71) levels. Similar testing on task completion time revealed a significantly lower time spent in Block 2 than in Block 1 for the easy (*p* < 0.001, d = 3.59), medium (*p* < 0.001, d = 2.89) and hard (*p* < 0.001, d = 1.37) levels. Post hoc testing between task difficulty levels of Block 1 indicated significant decreases in score from medium to hard levels (*p* = 0.023, d = 0.74), and completion time from easy to medium (*p* = 0.041, d = 0.64) and easy to hard (*p* = 0.012, d = 0.81). Alternatively, in Block 2, significant decreases in score were observed from easy to medium (*p* = 0.023, d = 0.71), medium to hard (*p* = 0.001, d = 1.05) and easy to hard (*p* < 0.001, d = 1.76), while significant increases in completion time were observed from medium to hard (*p* < 0.001, d = −1.35) and easy to hard (*p* < 0.001, d = −1.42).

fNIRS Results. The fNIRS measures in Oxy, HbO and HbR at the PFC region were analyzed by using changes in four regions: left DLPFC; left AMPFC; right AMPFC; right DLPFC. The main effect of block was significant in left DLPFC (HbO: χ^2^(1) = 15.47, *p* < 0.001; HbR: χ^2^(1) = 9.76, *p* = 0.002; Oxy: χ^2^(1) = 25.42, *p* < 0.001), left AMPFC (HbO: χ^2^(1) = 6.34, *p* = 0.012; HbR: χ^2^(1) = 8.13, *p* = 0.004; Oxy: χ^2^(1) = 12.82, *p* < 0.001) and right DLPFC (HbO: χ^2^(1) = 31.21, *p* < 0.001; HbR: χ^2^(1) = 16.07, *p* < 0.001; Oxy: χ^2^(1) = 42.53, *p* < 0.001). These results are also depicted in Figure 4. The main effect of task difficulty was significant primarily in left DLPFC (HbO: χ^2^(2) = 15.54, *p* < 0.001; HbR: χ^2^(2) = 6.19, *p* = 0.045; Oxy: χ^2^(1) = 10.66, *p* = 0.005). Post hoc testing between task difficulty levels indicated significant decreases in activity within left DLPFC from easy to medium (HbO: *p* < 0.001, d = 0.96; Oxy: *p* = 0.002, d = 0.82) and easy to hard (HbO: *p* = 0.005, d = 0.71; Oxy: *p* = 0.009, d = 0.70). The interaction between block and task difficulty was significant primarily in left DLPFC (HbO: χ^2^(2) = 8.71, *p* = 0.013; HbR: χ^2^(2) = 6.61, *p* = 0.037; Oxy: χ^2^(1) = 8.21, *p* = 0.016). Post hoc testing of the same difficulty levels across blocks revealed a significantly lower activity in Block 2 than in Block 1 for the easy (HbO: *p* < 0.001, d = 1.65; HbR: *p* = 0.002, d = −1.29; Oxy: *p* < 0.001, d = 1.93) level. Post hoc testing between task difficulty levels of Block 1 indicated significant decreases in activity from easy to medium (HbO: *p* < 0.001, d = 1.42; HbR: *p* = 0.006, d = −1.10; Oxy: *p* < 0.001, d = 1.45) and easy to hard (HbO: *p* < 0.001, d = 1.38; HbR: *p* = 0.020, d = −0.92; Oxy: *p* = 0.002, d = 1.20). In Block 2, no significant differences in activity between levels were observed.

In the second block, participants’ oxygen changes (OXY and HbO) in the PFC dropped for all task difficulty levels (Figure 5a,b), while their scores increased, as did their task completion time, which was the time spent to complete the tasks (Figure 4).

Additional analysis was performed on the individual fNIRS optodes. Within the scope of this study, we primarily focused on the left dorsolateral prefrontal cortex region, specifically optode-3 and optode-4, as shown in Figure 3b, as both optodes acquire hemodynamic changes from the PFC region, which is known to be associated with working memory and learning [15,25]. The topographic mapping and the relative oxygenation changes are depicted on the left dorsolateral prefrontal cortex (left DPFC) for each task block in Figure 6. Oxygenation changes in the left DPFC were significantly lower during the second training session (Block 2) compared to the initial training through Block 1 (*p* < 0.05).

At the individual spatial analysis, the differences in relative oxygen changes for the easy tasks of the two blocks were found to be statistically significant (*p* < 0.05) on optode-3. Similarly, there was a statistically significant difference between Block 1’s and Block 2’s oxygenation measurement for the easy and medium task difficulty levels (*p* < 0.05) on optode-4 (Figure 7).

## 4. Discussion

This study focused on investigating brain activity changes through fNIRS data analysis to understand the training effect of RAS modules and their relationship with the performance scores acquired from the simulation system. As it was reported in other studies, fNIRS data enable us to reveal the interaction between cortical regions dedicated to task execution during visuomotor learning [26]. In agreement with these studies, the results here also reveal that relative oxygenation changes in the prefrontal cortex were significantly lower during the second training session compared to the first session while the trainees were performing the tasks with varying difficulty levels. This finding was confirmed by the post hoc testing of the same difficulty levels across blocks (Table 2), which indicated significantly lower oxygenation in Block 2 than in Block 1 for the easy level (HbO: *p* < 0.001, d = 1.65; Oxy: *p* < 0.001, d = 1.93). We posit that this decrease in HbO and Oxy within Block 2 could be due to the lower demand at the corresponding attentional and working memory areas in response to increased task proficiency. Our findings are in line with prior studies which examined task practice and found decreases in the extent or intensity of activations, particularly in the attentional and control areas [27]. There is considerable evidence that practicing a task tends to be associated with overall lower brain activity in the prefrontal areas [28]. Hence, a significant decrease in oxygenation changes was expected when comparing Block 2 with Block 1. On the other hand, the increase in HbR suggests that, even though there could be a decrease in cerebral blood flow due to the lower demand, the local rate of oxygen consumption may still be elevated to stay on task, as verified by the increased task proficiency in Block 2. This result also reveals an association with an inverse bold response mechanism which has been observed in prior fMRI studies [29]. 

In support of these neurophysiological findings, the behavioral task performances of the participants were improved in the second block for all task difficulty levels. Statistically significant decreases in terms of task completion time for tasks with easy, medium and hard levels were also observed when task completion times in the first and second block were compared. In support of this finding, the post hoc testing of the same difficulty levels across blocks revealed a significantly higher score in Block 2 than in Block 1 for the easy (*p* < 0.001, d = −1.72) and medium (*p* = 0.023, d = −0.71) levels (Table 1). On the other hand, the same post hoc test of task completion time showed a significantly lower time spent in Block 2 than in Block 1 for the easy (*p* < 0.001, d = 3.59), medium (*p* < 0.001, d = 2.89) and hard (*p* < 0.001, d = 1.37) levels. 

Training Effect across Task Difficulty: We observed higher oxygenation and behavioral performance changes in the ‘easy’ condition. This may present conflict with other cognitive workload studies as the main hypothesis in such studies would be that there is a significant main effect between task difficulty and mental workload, as measured by Oxy [14,30]. That is, it is hypothesized to observe lower oxygenation in the easy condition compared to the difficult condition. However, there is an inconsistency here, where one possible explanation would be a lack of enough block designs for each difficulty condition. In this study, the scope was to investigate the training effect; hence, the experimental protocol aimed to elicit cortical changes in response to skill acquisition or familiarization. This is also supported by the behavioral score changes, and it is clearly observed that the trainees could have leveraged the first condition (easy) in order to become familiar with the task using a high level of mental engagement. This ‘easy’ condition was the first one and thus would have served as a training task for novice participants. This should also be considered as a limitation of this study, as task difficulty is a primary factor that can affect the development of neuroimaging-based biomarkers for an assessment of the level of expertise. To test the hypothesis related to the task difficulty and oxygenation changes in the PFC region, studies should include novice and expert participants. The participants in this study were all novice subjects, and this may also be another possible explanation of our mixed findings relative to workload levels. Shewokis et al. [31] studied novices, presenting different hemodynamic patterns, and also reported surgical simulation studies with optical brain imaging, which mainly found higher prefrontal activation across novice surgeons [32]. 

Advances in the measurement of neurophysiological biomarkers, such as brain activity changes, bring promising new assessment alternatives within reach for traditional training scenarios and modules. Currently, the most widely used brain activity measures are functional magnetic resonance imaging (fMRI) and electroencephalography (EEG), in addition to the technique described here, namely, fNIRS. fMRI is widely used to study the operational organization of the human brain and has demonstrated validity in mapping changes in brain hemodynamics produced by human mental tasks [33]. However, use of fMRI in field operations is limited due to the restrictions imposed on participants regarding movement. EEG measures of cognitive workload have been reported in a broad range of studies and provide a high temporal resolution compared to fNIRS and fMRI [34]. 

Both behavioral and neurophysiological measurement methods in practical training programs will help to assess operational readiness, which is one of the overarching goals for safety-critical task training. Quantitative assessments of joint human–system performance in military and clinical settings, such as integration of autonomous systems or new VR-based training, or designing training curricula derived from measures of expertise development through neural efficiency, are just a few examples illustrating the potential role of these methods. For instance, previous studies posited that the systems assessing brain activation during surgical tasks may be used as assessment methods for endoscopic surgical skills [35,36]. As suggested in similar studies using fNIRS systems, this study also suggests utilizing brain-based measures as an additional monitoring tool for assessing the training effect of robotic surgery simulators via hemodynamic changes in the prefrontal cortex regions of trainees during robotic surgery simulation sessions [37,38]. Future studies should include expert surgeons as a control group, and further blocks with different difficulty levels which should be designed and administered in a random order. Furthermore, one of the limitations of this study is the lack of short source–detector pairs which would allow fNIRS measures from the superficial layers. fNIRS signals could be confounded by systemic changes due to the heart, respiratory, myogenic and scalp blood flow [39]. Hence, use of fNIRS sensors with capability of multi-distance measures can help to tease out these confounding factors, enabling one to reliably distinguish neural changes elicited by cognitive stimuli.

## 5. Conclusions

Neurophysiological measures from the prefrontal cortex can serve as a performance-monitoring metric while a trainee surgeon moves from novice to expert. This study showed significant differences in PFC activation during basic skill acquisition via a high-fidelity Da Vinci surgical system, suggesting that they can be used as an additional assessment tool. Our work reports a correlation analysis with the standard behavioral performance metrics and brain activity changes. The findings suggest that integration of neurophysiological measures, such as hemodynamic changes assessed by wearable fNIRS sensors, with a built-in performance-tracking system in the training simulator, such as task scores and completion time, would enable a reliable performance assessment of joint human and training systems in clinical skill acquisition.

## Figures and Tables

**Figure 1 brainsci-11-00937-f001:**
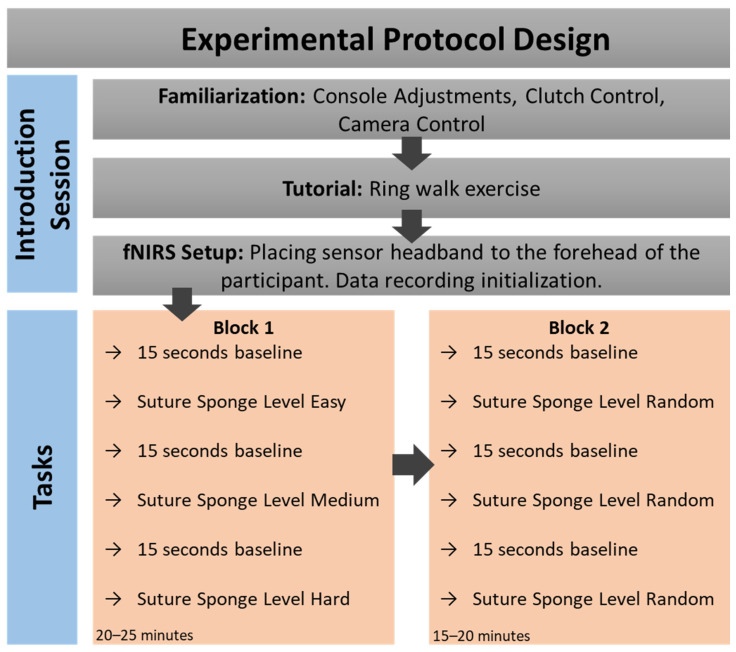
Experimental protocol sessions, timeline and detailed description of the steps in each block.

**Figure 2 brainsci-11-00937-f002:**
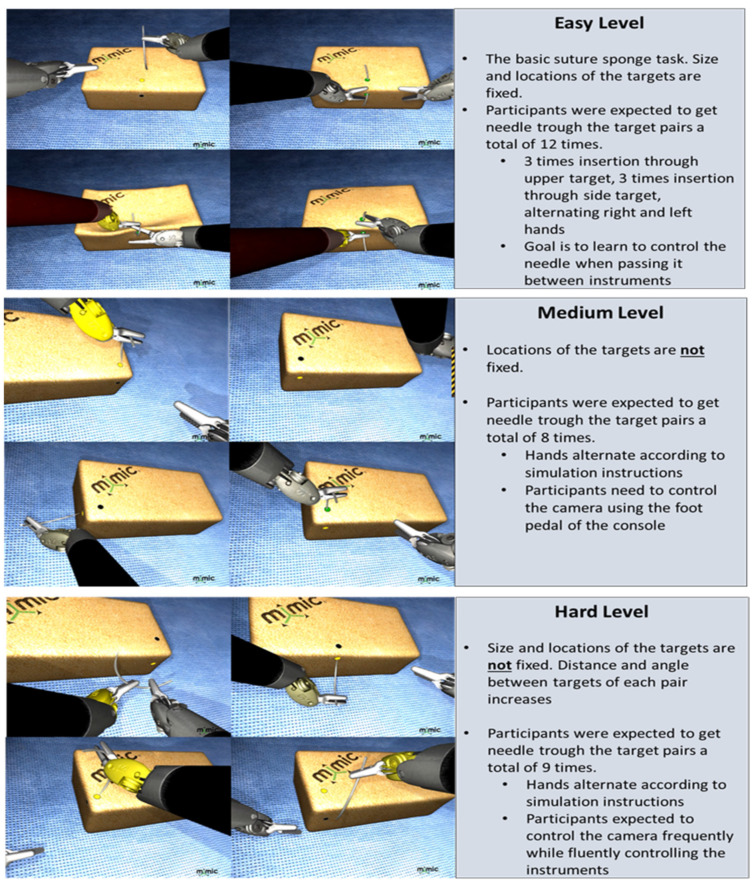
Suture sponge task difficulty levels include easy, medium and hard levels. Detailed descriptions for each level are provided to illustrate examples of standard training tasks as part of a professional simulator (dV-Trainer^®^, Mimic Technologies, Inc. Seattle, WA, USA).

**Figure 3 brainsci-11-00937-f003:**
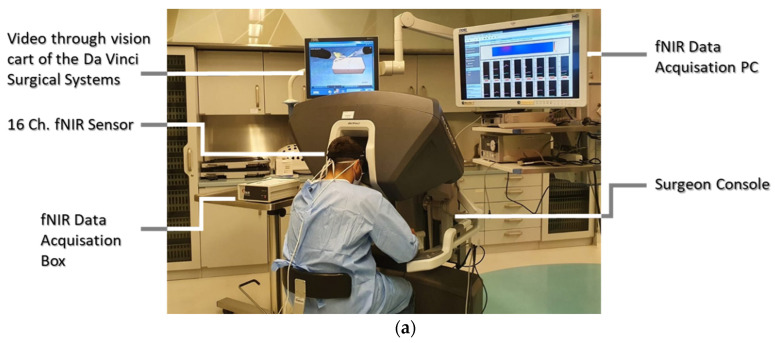
(**a**) A participant performing a task while the fNIRS headband acquires the data; (**b**) 16-optode fNIRS sensor configuration depicted on the PFC region covering the left and right hemispheres.

**Figure 4 brainsci-11-00937-f004:**
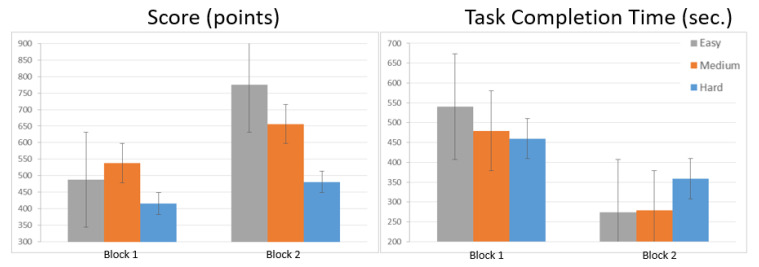
Score and task completion time: comparison of measurements between blocks for simulator performance data with error of the mean.

**Figure 5 brainsci-11-00937-f005:**
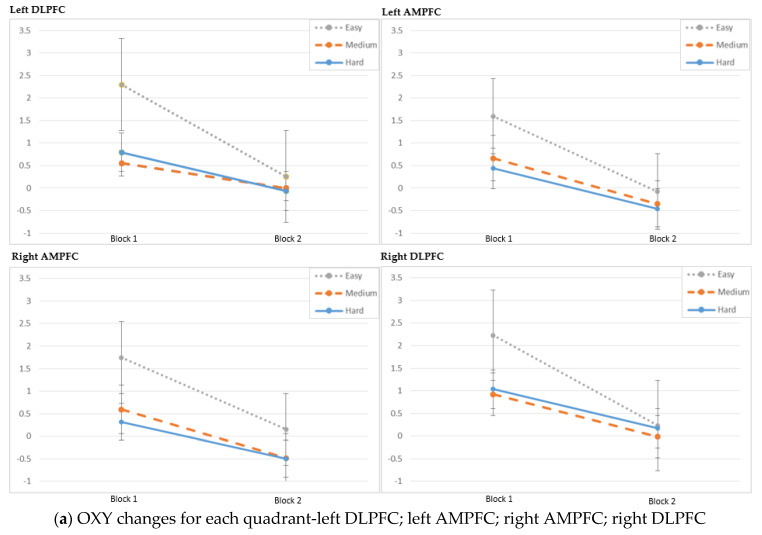
(**a**) Oxygenation changes (OXY); (**b**) oxygenated hemoglobin (HbO) changes; (**c**) deoxygenated hemoglobin (HbR) changes with error of the mean.

**Figure 6 brainsci-11-00937-f006:**
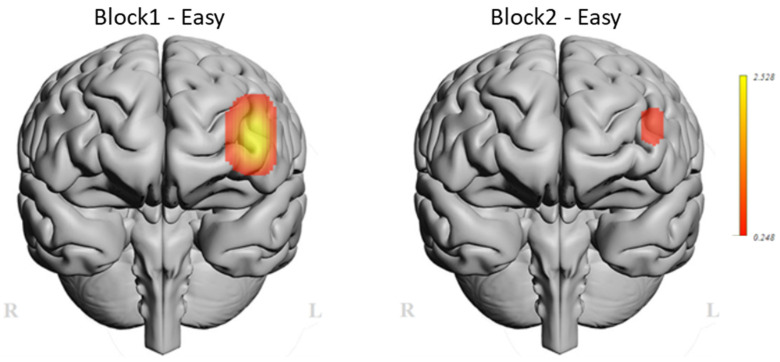
Oxygenation differences (Oxy) from left PFC between Block 1 and Block 2 initial tasks.

**Figure 7 brainsci-11-00937-f007:**
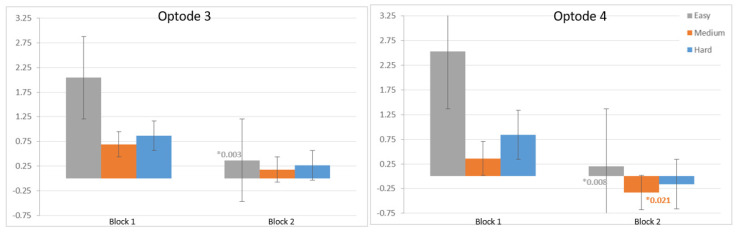
Decrease in oxygenation from the measures on optode-3 and optode-4 for each level between the blocks, with error of the mean bars.

**Table 1 brainsci-11-00937-t001:** Behavioral measure LME model and log-likelihood test results.

Dependent Variable	Term	logLik	Chisq (χ^2^)	df	*p*-Value
Score	Block	−982.33	21.92	1.00	0.000
Score	Task Difficulty	−968.57	27.52	2.00	0.000
Score	Block:Difficulty	−962.88	11.37	2.00	0.003
Completion Time	Block	119.03	114.37	1.00	0.000
Completion Time	Task Difficulty	122.38	6.70	2.00	0.035
Completion Time	Block:Difficulty	136.71	28.68	2.00	0.000

**Table 2 brainsci-11-00937-t002:** fNIRS measure (Oxy, HbO, HbR) LME model and log-likelihood test results.

Dependent Variable	Term	logLik	Chisq (χ^2^)		df	*p*-Value
**OXY**						
Left DLPFC	Block	−165.83	25.42		1.00	0.000
Left DLPFC	Task Difficulty	−160.50	10.66		2.00	0.005
Left DLPFC	Block:Difficulty	−156.39	8.21		2.00	0.016
Left AMPFC	Block	−177.53	12.82		1.00	0.000
Left AMPFC	Task Difficulty	−174.02	7.01		2.00	0.030
Left AMPFC	Block:Difficulty	−171.87	4.30		2.00	0.116
Right AMPFC	Block	−192.93	17.20		1.00	0.000
Right AMPFC	Task Difficulty	−183.12	19.63		2.00	0.000
Right AMPFC	Block:Difficulty	−181.97	2.31		2.00	0.316
Right DLPFC	Block	−180.78	42.53		1.00	0.000
Right DLPFC	Task Difficulty	−173.71	14.14		2.00	0.001
Right DLPFC	Block:Difficulty	−169.11	9.21		2.00	0.010
**HbO**						
Left DLPFC	Block	−161.33	15.47	1.00		0.000
Left DLPFC	Task Difficulty	−153.56	15.54	2.00		0.000
Left DLPFC	Block:Difficulty	−149.21	8.71	2.00		0.013
Left AMPFC	Block	−197.71	6.34	1.00		0.012
Left AMPFC	Task Difficulty	−195.44	4.54	2.00		0.103
Left AMPFC	Block:Difficulty	−194.42	2.05	2.00		0.359
Right AMPFC	Block	−202.85	9.39	1.00		0.002
Right AMPFC	Task Difficulty	−202.20	1.29	2.00		0.526
Right AMPFC	Block:Difficulty	−200.13	4.16	2.00		0.125
Right DLPFC	Block	−173.56	31.21	1.00		0.000
Right DLPFC	Task Difficulty	−168.14	10.84	2.00		0.004
Right DLPFC	Block:Difficulty	−163.48	9.32	2.00		0.009
**HbR**						
Left DLPFC	Block	−124.97	9.76	1.00		0.002
Left DLPFC	Task Difficulty	−121.88	6.19	2.00		0.045
Left DLPFC	Block:Difficulty	−118.57	6.61	2.00		0.037
Left AMPFC	Block	−144.26	8.13	1.00		0.004
Left AMPFC	Task Difficulty	−144.18	0.15	2.00		0.929
Left AMPFC	Block:Difficulty	−141.22	5.93	2.00		0.052
Right AMPFC	Block	−156.38	1.71	1.00		0.191
Right AMPFC	Task Difficulty	−153.64	5.47	2.00		0.065
Right AMPFC	Block:Difficulty	−150.83	5.62	2.00		0.060
Right DLPFC	Block	−142.37	16.07	1.00		0.000
Right DLPFC	Task Difficulty	−139.50	5.73	2.00		0.057
Right DLPFC	Block:Difficulty	−136.88	5.26	2.00		0.072

## Data Availability

Data sharing is not applicable to this article due to privacy restrictions.

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
