# Peer review of "Studying Brain Activation during Skill Acquisition via Robot-Assisted Surgery Training"

_brainsci, 2021, doi:10.3390/brainsci11070937_

Round 1
Reviewer 1 Report
Interesting paper, few suggestions/comments:
1- What are the possible application of this method? Please add to discussion
2- The method can differentiate easy versus more complicated task. What about the technical proficiency? can it compare good vs poor performance?
3- Why didn't the authors consider real time surgeries or even using tissue or inanimate model rather than simulator?
4- Any discomfort reported by participants?
5- How does this method compare with EEG and functional MRI? please elaborate in the discussion.
6- Any subjective feedback from participants?
Author Response
We would like to thank the Reviewer for the detailed evaluation of our paper and for the valuable feedback. We addressed each comment and suggestion of the Reviewer in the attached response letter. We provided the significantly revised version of the manuscript with ‘track changes’. We believe the paper is improved in terms of its clarity, content, analysis and discussions. We hope that the reviewer find the revisions satisfactory.

Reviewer 2 Report
In this article, the authors investigated changes in brain activity in resident surgeons by measuring hemodynamic changes from the prefrontal cortex using fNIRS while the residents completed RAS training. While the study in interesting, I have some major and minor concerns regarding the data acquisition and analysis.
- One of my main major comments has to do with the lack of short channels during data acquisition. fNIRS is inherently sensitive to systemic physiology and currently short channels are becoming the standard for regressing out extracerebral contributions. Without performing this step, the results are not accurate as systemic physiology can lead to false positives and negatives (see Tachtsidis, I., & Scholkmann, F. (2016). False positives and false negatives in functional near-infrared spectroscopy: issues, challenges, and the way forward. Neurophotonics, 3(3), 031405. https://doi.org/10.1117/1.NPh.3.3.031405). While the authors preprocessed the data by filtering this is not sufficient. Another concern is the sampling frequency used. 2Hz is not sufficient to capture heart rate in all participants and the lack of high sampling frequency can lead to aliasing and the inability to filter our heart rate.
- Instead of relying on calculating the mean/median change in the signal, is it possible to run a general linear model with a block design convolved with an hrf? I understand the length of blocks is not constant, but you can take that into account. GLM will be more robust than looking at the median/mean.
- Since short channels are not available, it is important to take both oxy and deoxyhemoglobin into account in the data analysis. This reduces the instances of false positives. Please redo the analysis taking both chromophores into account.
- Please plot the average change in concentration of oxy and deoxyhemoglobin across participants with the appropriate error bars. It is important to visualize the time courses to determine the presence of a hemodynamic response.
- Please add the individual results to the tables. This could be in the supplementary material.
- Last sentence of the abstract is confusing. Please reword.
- Where appropriate, please add units to the table (example: the values in table 3 are in seconds?).
- Figure 4 and 6: please add error bars to represent changes across participants.
- Line 206-208: Sentence very confusing. Please reword.
- For some values of oxyhemoglobin in the second block it looks like oxyhemoglobin is decreasing below the baseline level. What does this signify? Deactivation or inverse oxygenation? Please add some discussion.
Author Response
We would like to thank the Reviewer for the detailed evaluation of our paper and for the valuable feedback. We addressed each comment and suggestion of the Reviewer in the attached letter. We provided the significantly revised version of the manuscript with ‘track changes’. We believe the paper is improved in terms of its clarity, content, analysis and discussions. We hope that the reviewer find the revisions satisfactory.

Round 2
Reviewer 2 Report
I would like to thank the authors for making the suggested changes to the manuscript. I believe this has improved the quality of the manuscript. I still however have some questions that I am hoping to be answered:
1- I am confused by Figure 5. Why would oxy decrease for the second block and deoxy increase? This suggests deactivation or inverse oxygenation. Not sure why this would happen (please see Abdalmalak et al. Neuroscience Letters, 2020 for more info on inverse oxygenation). Some discussion is needed as to why this would happen.
2- I would still like to see the average time courses (i.e. average change in oxy and deoxy plotted in the same figure) to visualize the hemodynamic response. This could help explain my first comment. You could plot the average time course per channel averaged across participants.
Author Response
We would like to thank again the Reviewer for further evaluation and feedback. We addressed each comment and suggestion of the Reviewer in this second response letter, and uploaded revised manuscript.
